# Degradation of *Gadd45* mRNA by nonsense-mediated decay is essential for viability

Jonathan O Nelson[1], Kristin A Moore[2,3], Alex Chapin[1], Julie Hollien[2,3], Mark M Metzstein[1]*

[1]Department of Human Genetics, University of Utah, Salt Lake City, United States; [2]Department of Biology, University of Utah, Salt Lake City, United States; [3]Center for Cell and Genome Sciences, University of Utah, Salt Lake City, United States

**Abstract** The nonsense-mediated mRNA decay (NMD) pathway functions to degrade both abnormal and wild-type mRNAs. NMD is essential for viability in most organisms, but the molecular basis for this requirement is unknown. Here we show that a single, conserved NMD target, the mRNA coding for the stress response factor growth arrest and DNA-damage inducible 45 (GADD45) can account for lethality in *Drosophila* lacking core NMD genes. Moreover, depletion of *Gadd45* in mammalian cells rescues the cell survival defects associated with NMD knockdown. Our findings demonstrate that degradation of *Gadd45* mRNA is the essential NMD function and, surprisingly, that the surveillance of abnormal mRNAs by this pathway is not necessarily required for viability.

## Introduction

Maintaining proper gene expression is critical for normal development and physiology. In addition to *de novo* transcription, mRNA stability substantially contributes to forming the landscape of expression in a cell. The nonsense-mediated mRNA decay (NMD) pathway is a *trans*-acting mechanism that destabilizes mRNAs, and is best known for its well-described role as a quality control system, degrading abnormal mRNAs containing premature termination codons (PTCs) (*Celik et al., 2015*). NMD also degrades many wild-type endogenous mRNAs and thus is an important aspect of their post-transcriptional (*Peccarelli and Kebaara, 2014*). Loss of either of the core NMD genes *Upf1 (Rent1)* or *Upf2* causes lethality in most eukaryotes (*Kerényi et al., 2008*; *Medghalchi et al., 2001*; *Metzstein and Krasnow, 2006*; *Weischenfeldt et al., 2008*; *Wittkopp et al., 2009*), indicating regulation of mRNA stability by NMD is critical for viability. However, the relative contributions to lethality from ectopic stabilization of PTC-containing mRNAs or endogenous NMD targets in NMD mutants remains unclear (*Hwang and Maquat, 2011*).

To identify which ectopically stabilized mRNAs are responsible for inducing lethality in NMD mutants, we performed an unbiased genetic suppressor screen seeking to restore viability in a *Drosophila* NMD mutant. To detect subtle increases in survival, we screened to suppress the lethality of animals mutant for the partially viable, hypomorphic *Upf2^{25G}* allele, of which 10% survive to adulthood (*Chapin et al., 2014*; *Metzstein and Krasnow, 2006*). We crossed this allele to heterozygous deficiencies to simultaneously reduce the mRNA abundance of several loci (*Figure 1A*). Of the 376 deficiencies tested, covering more than half the genome, ~10% suppressed NMD mutant lethality (*Figure 1B*, *Figure 1—figure supplement 1A*). The suppression effect could not be explained by a reduction in overall mRNA load, as there was only a weak correlation between the increase in mRNAs expressed from a genomic region upon loss of NMD function and the strength of

*For correspondence: markm@genetics.utah.edu

**Competing interests:** The authors declare that no competing interests exist.

**eLife digest** Messenger RNA (mRNA) molecules act as the templates from which proteins are made, and so control the amount of protein in a cell. Having too much of certain proteins can harm cells. Additionally, some mRNAs contain errors, and so can create faulty proteins that may also harm the cell.

Cells have therefore developed ways to destroy excess or error-ridden mRNAs to avoid a deadly build up of proteins. One such quality control mechanism is called nonsense-mediated decay (NMD). This mechanism is so important that cells that cannot perform nonsense-mediated decay die, although it is not clear exactly what kills the cells.

Now, Nelson et al. have found that fruit flies whose cells are unable to perform nonsense-mediated decay die because a harmful protein called Gadd45 builds up in the cells. In normal cells, nonsense-mediated decay destroys the mRNA that relays the instructions for making Gadd45, which keeps the amount of the Gadd45 protein in the cell low. Further experiments show that removing Gadd45 from cells that lack nonsense-mediated decay saves the flies. Removing Gadd45 from human and mouse cells that are unable to perform nonsense-mediated decay also allows these cells to survive.

These findings imply that the only nonsense-mediated decay function needed for cells to live is the destruction of *Gadd45* mRNA. This further implies that most faulty and normal mRNAs that are normally destroyed by nonsense-mediated decay do not cause the cells to die when nonsense-mediated decay is lost.

Learning that creating faulty proteins when nonsense-mediated decay is lost is not necessarily harmful to cells opens new possibilities to treating numerous genetic diseases. In some diseases, cells can only produce faulty forms of a particular protein. Nonsense-mediated decay normally destroys all of these mutant proteins, but it may sometimes be better to have faulty versions of a protein than to have none of it. Safely getting rid of nonsense-mediated decay by also eliminating Gadd45 from cells may therefore be a treatment strategy worth exploring.

suppression when that region was removed by a deficiency (*Figure 1—figure supplement 1B*). Rather, deficiencies that suppressed NMD-mutant lethality clustered in three genomic regions (*Figure 1—figure supplement 1A*). These findings suggest that NMD mutant lethality is not the result of a global excess of nonspecific mRNAs, but rather is mediated by specific genes residing within the few identified regions.

We expected that any specific genes mediating NMD-mutant lethality would have increased expression levels in an NMD mutant and be a direct NMD target. The only gene located within the suppressing regions to fit these criteria is *Gadd45* (*Figure 1C*, *Figure 1—figure supplement 2A–C*) (*Chapin et al., 2014*). To determine if NMD targeting of *Gadd45* mRNA is critical for viability, we generated a *Gadd45* null allele, *F17*, which completely removes the *Gadd45* coding region (*Figure 1—figure supplement 3A*) and eliminates *Gadd45* mRNA expression (*Figure 1—figure supplement 2A*). As a heterozygote, $Gadd45^{F17}$ suppressed $Upf2^{25G}$lethality as strongly as the corresponding deficiency identified by our screen (*Figure 1D*). We found that $Gadd45^{F17}$ homozygous mutants are fully viable (*Figure 1—figure supplement 3B*), allowing us to test complete loss of *Gadd45* for the suppression of NMD-mutant lethality. Homozygous $Gadd45^{F17}$ restored full viability to $Upf2^{25G}$mutants, and remarkably even partially suppressed the complete lethality observed in null *Upf1* and *Upf2* mutants (*Frizzell et al., 2012*; *Metzstein and Krasnow, 2006*) (*Figure 1D*). Importantly, neither reducing nor eliminating *Gadd45* restored NMD function to $Upf2^{25G}$ mutants, as measured by the expression of both an endogenous NMD target (*Figure 1—figure supplement 4A*) and PTC-containing mRNAs (*Figure 1—figure supplement 4B*).

In mammals, GADD45 activates the MTK1/MEKK4 kinase in a well-defined stress response pathway (*Takekawa and Saito, 1998*). Strikingly, the *Drosophila* MTK1 orthologue, *Mekk1*, resides within another $Upf2^{25G}$ suppressing region (*Figure 1E*). Similar to *Gadd45*, we found that *Mekk1* null mutants (*Inoue et al., 2001*) suppressed *Upf1* and *Upf2* mutant lethality (*Figure 1F*). This suppression was not as strong as that caused by a loss of *Gadd45*, revealing that although MEKK1 mediates

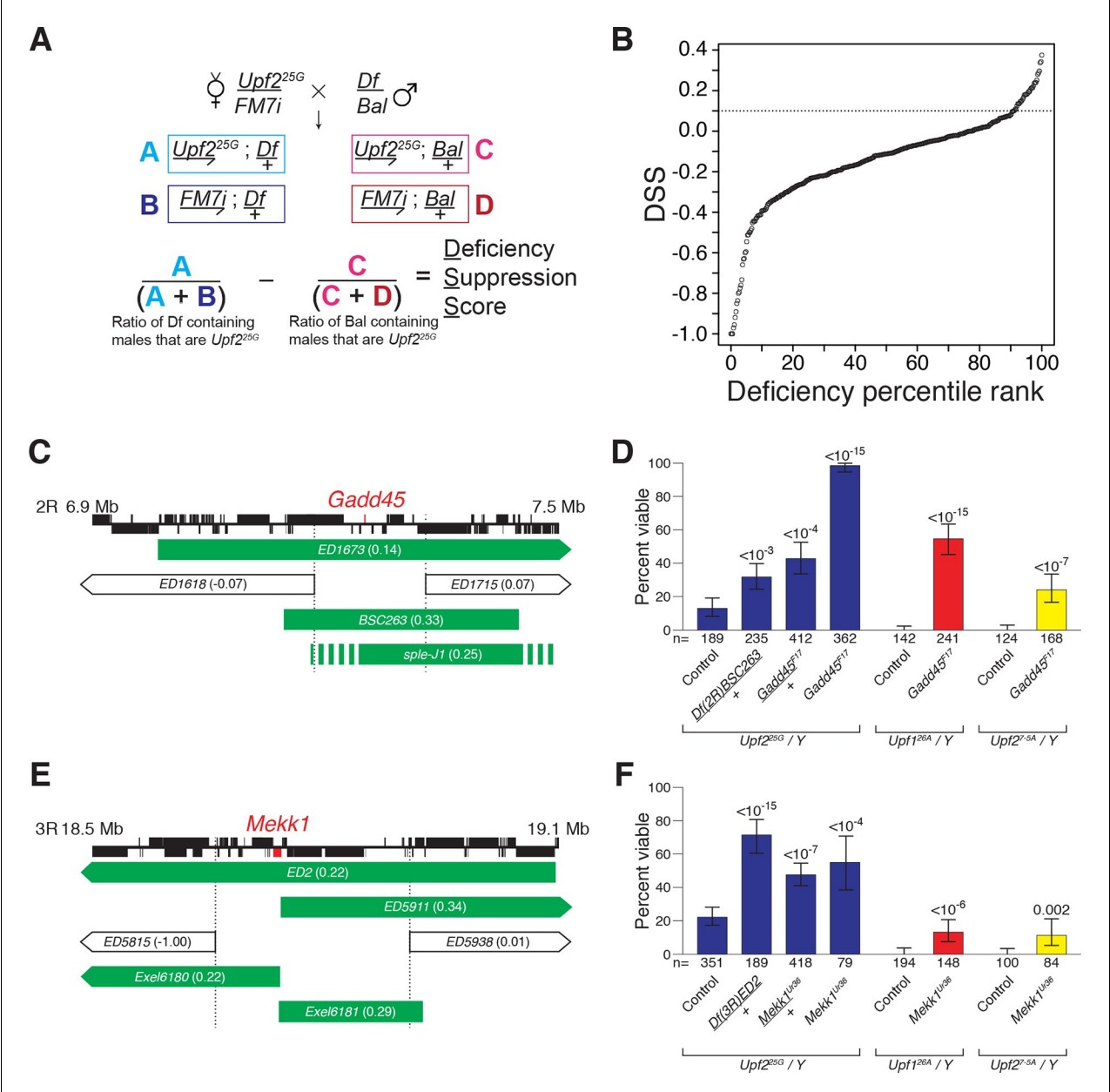

**Figure 1.** *Drosophila* suppressor screen identifies the *Gadd45* pathway as the inducer of NMD-mutant lethality. (**A**) Scheme to screen deficiencies for the suppression of $Upf2^{25G}$ partial lethality. The Deficiency Suppression Score (DSS) represents the relative difference in $Upf2^{25G}$ viability when crossed to a heterozygous deficiency (*Df*) compared to when crossed to a balancer (*Bal*) (See Methods). (**B**) DSS from 376 screened deficiencies ranked by score. A DSS greater than 0.1 (dotted line) indicates that deficiency suppresses $Upf2^{25G}$ lethality. (**C** and **E**) Candidate suppressing regions uncovering *Gadd45* (**C**) and *Mekk1* (**E**). DSSs are shown in parenthesis. Dotted lines denote extent of regions deleted by suppressing deficiencies but not non-suppressing deficiencies. Filled blocks on chromosomes indicate predicted gene spans, *Gadd45* pathway genes are indicated in red; suppressing deficiencies indicated in green, *sple-J1* has undefined breakpoints located within hashed regions. (**D** and **F**) NMD mutant adult viability in combination with $Gadd45^{F17}$ (**D**) or $Mekk1^{Ur36}$ (**F**) mutants. $Upf1^{26A}$ and $Upf2^{7-5A}$ are null alleles (**Frizzell et al., 2012**; **Metzstein and Krasnow, 2006**). p-value compared to controls determined by the test of equal or given proportions indicated. Error bars represent 95% confidence interval of the binomial distribution. n equals total number of animals scored in each cross.

The following figure supplements are available for figure 1:

**Figure supplement 1.** Reduced expression of specific loci, not overall mRNA abundance, produces NMD mutant suppression by deficiencies.

**Figure supplement 2.** Drosophila Gadd45 is an endogenous direct NMD target.

*Figure 1 continued on next page*

*Figure 1 continued*

**Figure supplement 3.** F17 is a null allele of Gadd45.

**Figure supplement 4.** Loss of Gadd45 does not restore NMD activity in NMD mutants.

NMD mutant lethality, it is likely that GADD45 has additional downstream effectors that influence viability. Overall, our findings reveal that increased *Gadd45* mRNA stability is the major factor inducing NMD mutant lethality, primarily via increased MEKK1 activity.

Activation of MTK1 in mammals triggers a MAPK signaling cascade that promotes apoptosis (*Takekawa and Saito, 1998*). Over-expression of *Gadd45* in *Drosophila* also induces apoptosis (*Peretz et al., 2007*). Interestingly, *Drosophila* cells lacking NMD function show excess cell death in a variety of tissues (*Avery et al., 2011*; *Frizzell et al., 2012*; *Metzstein and Krasnow, 2006*). To test if increased *Gadd45* contributes to this excess death, we used TUNEL staining to examine cell death in wing imaginal discs from $Upf2^{25G}$ mutant third instar larvae. This analysis revealed elevated levels of cell death compared to controls (*Figure 2A, B, E*), and this defect was completely suppressed by $Gadd45^{F17}$ (*Figure 2C–E*). To confirm that, this effect was not specific to the *Upf2* gene or *25G* allele, we examined the wing discs in mutants of another essential NMD gene, *Smg5*. We found that *Smg5* discs also showed elevated TUNEL signal, which was eliminated by loss of *Gadd45* (*Figure 2—figure supplement 1A–E*). These results demonstrate that excess *Gadd45* accounts for ectopic cell death in NMD mutant tissues.

To test if *Gadd45*-induced cell death is the only cellular defect in NMD mutants, we examined NMD function in the developing eye. NMD is required for proper development of eye cells, as clonal patches of NMD mutant cells in eyes are reduced in size (*Frizzell et al., 2012*; *Metzstein and Krasnow, 2006*). We found that *Gadd45* is partially responsible for this defect, as the size of eye-cell clones lacking NMD activity in a $Gadd45^{F17}$ background was increased, although not fully restored (*Figure 2F–J*). These results indicate that some, but not all, defects associated with loss of NMD are dependent on *Gadd45*.

*Gadd45* is one of the few genes that is directly regulated by NMD in both flies and mammals (*Huang et al., 2011*; *Tani et al., 2012*; *Viegas et al., 2007*), raising the possibility that excess *Gadd45* abundance may also contribute to the NMD-mutant lethality observed in mammalian cells (*Azzalin and Lingner, 2006*; *Li et al., 2015*; *Medghalchi et al., 2001*; *Weischenfeldt et al., 2008*). To test this hypothesis, we analyzed the effects of *Gadd45* and *Upf1* depletion in mouse NIH-3T3 cells. *Gadd45b* mRNA (also known as *MyD118*), which is expressed at least 10-fold higher than any other *Gadd45* paralogue in these cells (*Yue et al., 2014*), was degraded rapidly in a partially *Upf1*-dependent manner after transcription was blocked with actinomycin D (*Figure 3A*), and had increased expression during *Upf1* knockdown (*Figure 3D*), confirming it is sensitive to NMD. We found that transfection of 3T3 cells with siRNAs targeting *Upf1* resulted in significant reduction in cell counts after 48 hr (*Figure 3B*), but co-transfection with siRNAs targeting both *Upf1* and *Gadd45b* largely reversed this effect (*Figure 3B*). The reduction in cell counts was primarily due to increased cell death, as we found that ~25% of cells transfected with *Upf1* siRNA were undergoing apoptosis (*Figure 3C*). Co-transfection of siRNA targeting *Gadd45b* almost entirely eliminated this increase (*Figure 3C*), indicating the excess apoptosis observed in *Upf1*-knockdown cells was mostly due to increased *Gadd45* activity. However, while *Gadd45b* knockdown very greatly suppresses this excess death, it does not as fully rescue cell numbers, suggesting loss of NMD may lead to both Gadd45b-dependent cell death as well as a Gadd45b-independent effect on proliferation. This mirrors the conclusions we made about the partial suppression of cell number defects in the *Drosophila* eye. Importantly, *Upf1* mRNA expression was equivalently reduced and the expression of the mammalian endogenous NMD targets *Rassf1* and *CRCP* (*Tani et al., 2012*) was equivalently increased in both the single and double knockdown experiments (*Figure 3D*), indicating that the restoration of viability was not due to a recovery of NMD pathway activity.

To extend our analysis to other mammalian cells, we analyzed the role of *Gadd45* mediating the effects of loss of NMD in HEK293 cells. We found, similarly to 3T3 cells, that siRNA knockdown of *UPF1* in HEK293 cells led to increased *GADD45A* expression and reduced cell numbers compared

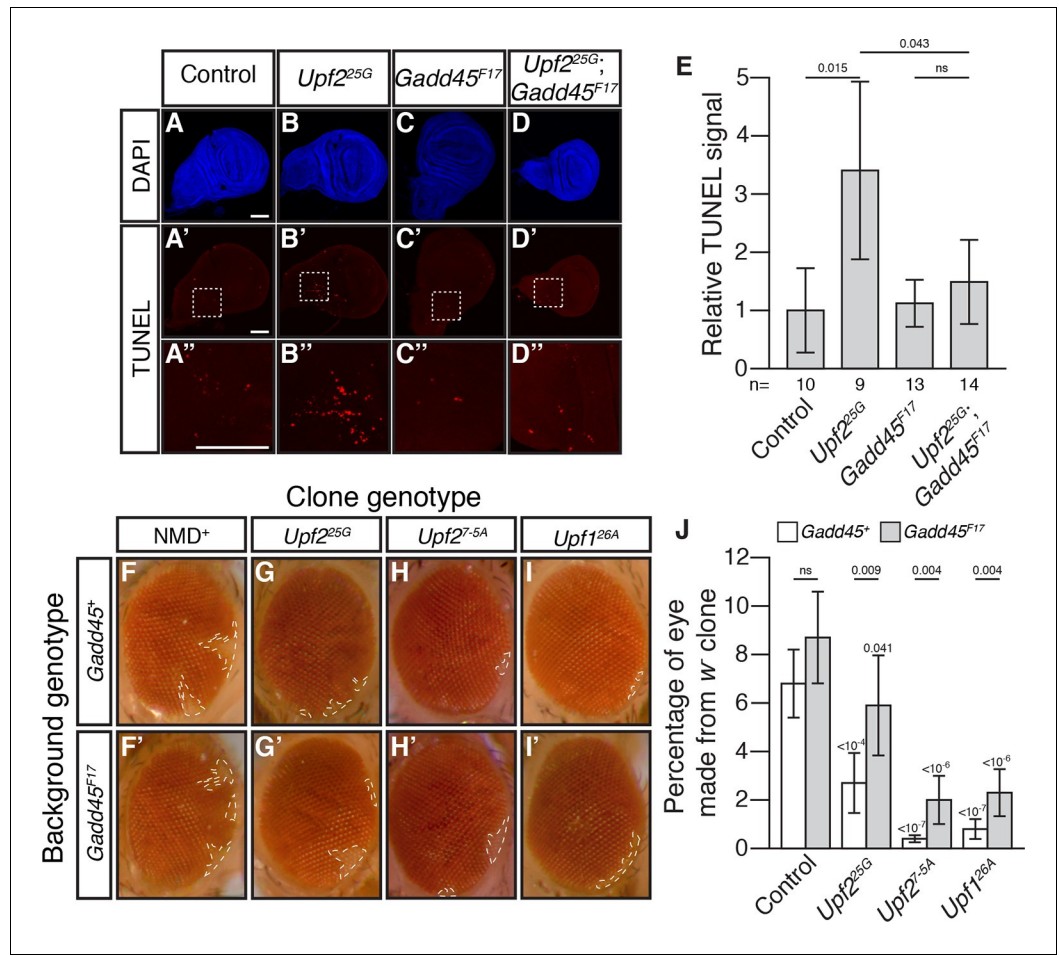

**Figure 2.** Loss of Gadd45 suppresses NMD-mutant cell death. (A to D) DAPI (blue) and (A'to D') TUNEL (red) staining in late 3<sup>rd</sup> instar larval wing discs from control (A); *Upf2^25G* (B); *Gadd45^F17* (C); and *Upf2^25G*; *Gadd45^F17* (D) animals. (A'' to D'') are 4x view of outlined section at the base of the blade of the wing disc from A'-D', respectively. Scale bar represents 100 μm. (E) Relative TUNEL signal in control and mutant wing discs, normalized to control. p-value between indicated samples using a two-sided Student's t-test are displayed. ns indicates a p-value greater than 0.05. Error bars represent 2 SEM. n equals total number of discs scored. (F to I) *w⁻* eye clones in *Gadd45⁺* and *Gadd45^F17* backgrounds. Dashed lines indicate clone boundaries. (J) Quantification of the fraction of the eye composed of *w⁻* cells in control and mutant eyes. p-values indicate differences between *Gadd45* mutant and control in the same NMD background (indicated by horizontal bars) or NMD mutant and control in the same *Gadd45* background (indicated by value above each individual bar), using a two-sided Student's t-test. ns indicates a p-value greater than 0.05. Error bars represent 2 SEM. n = 20 eyes for all conditions.

The following figure supplement is available for figure 2:

**Figure supplement 1.** Loss of *Gadd45* suppresses ectopic cell death in *Smg5* mutant wing discs.

to control siRNA (*Figure 3—figure supplement 1A,B*). Although transfection of siRNA targeting *GADD45A* alone slightly reduced HEK293 cell numbers, co-transfection with *UPF1* siRNA did not further reduce cell count (*Figure 3—figure supplement 1B*), and *UPF1* expression was equivalently reduced in the single and double knockdown conditions (*Figure 3—figure supplement 1C*). These results suggest that *UPF1* knockdown is no longer detrimental to HEK293 cell viability in the absence of *GADD45A* expression. We conclude that increased expression of mammalian *Gadd45* genes contributes to lethality in NMD-deficient mouse and human cells, as *Gadd45* does in *Drosophila*.

Deconvoluting the contributions to organismal viability of the PTC-surveillance versus gene-regulatory functions of NMD has been historically difficult (*Hwang and Maquat, 2011*). Here, we show that viability can be restored to *Drosophila* lacking core NMD factors when a single endogenous

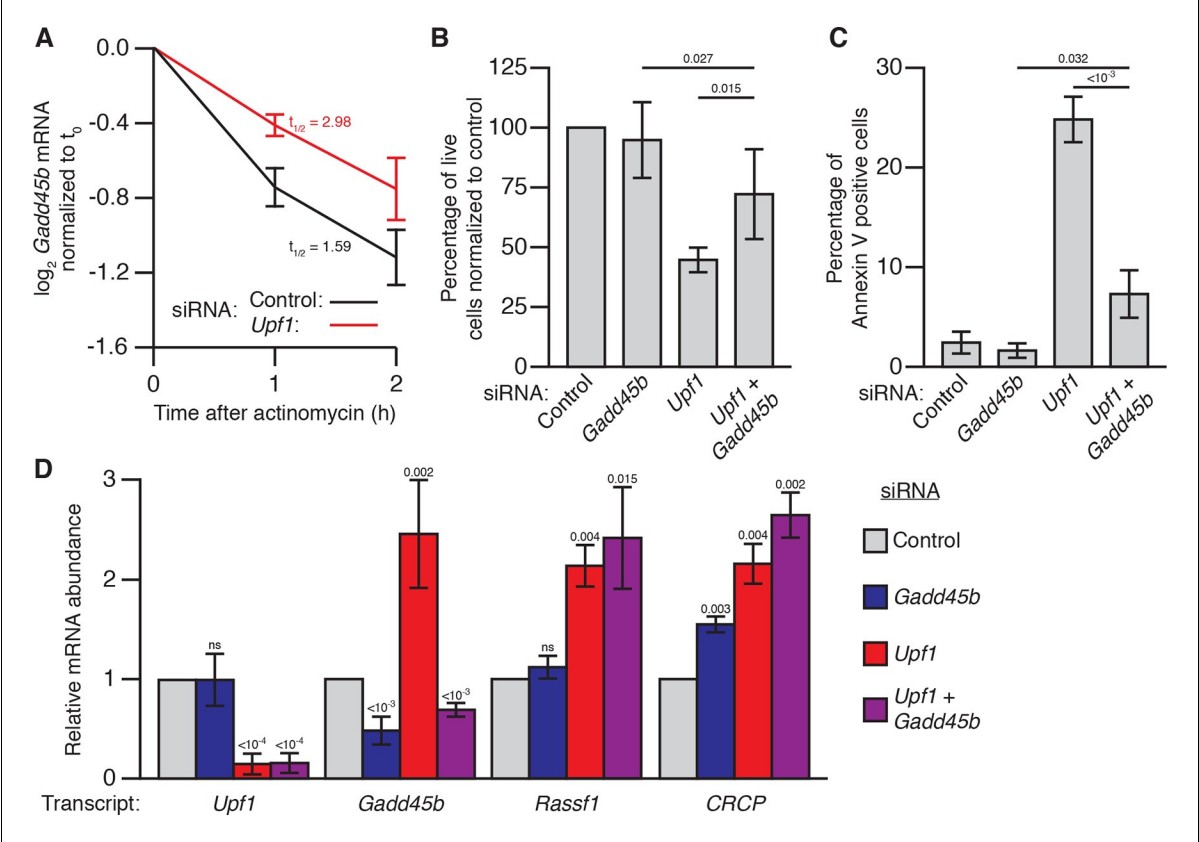

**Figure 3.** Gadd45b mediates cell lethality in Upf1 siRNA knockdown 3T3 mouse embryonic fibroblasts. (**A**) Relative *Gadd45b* mRNA expression measured by qRT-PCR in NIH-3T3 cells after 48 hr of control (black) or *Upf1* (red) siRNA treatment and 0 to 2 hr of actinomycin D treatment, normalized to expression prior to actinomycin treatment. The half-life calculated for each decay curve is indicated. (**B**) Relative viable cell count of *Upf1* and *Gadd45b* single and double siRNA treatment normalized to control siRNA. p-values display two-sided Student's t-test between indicated conditions. (**C**) Quantification of apoptosis as measured by annexin V staining. p-values display two-sided Student's t-test between indicated conditions. (**D**) Relative mRNA expression of *Upf1*, *Gadd45b*, and two mammalian endogenous NMD targets, *Rassf1* and *CRCP* (*Tani et al., 2012*) measured by qRT-PCR in *Gadd45b* and *Upf1* single and double siRNA knockdown cells, normalized to expression in the control siRNA condition. p-values display one-sided Student's t-test for each condition compared to control. Error bars represent 2 SEM.

The following figure supplement is available for figure 3:

**Figure supplement 1.** *GADD45A* mediates cell lethality in *Upf1* knockdown HEK293 cells.

NMD target, *Gadd45*, is eliminated, and that the requirement for the regulation of *Gadd45* by NMD is evolutionarily conserved from flies to mammals. Although our data suggest that up-regulation of *Gadd45* is a major factor contributing to lethality when NMD activity is lost, it is likely that other NMD targets also contribute to the observed lethality. In particular, viability is not restored to 100% in null *Upf1*; or *Upf2*; *Gadd45* double mutants. In addition, loss of *Gadd45* suppresses programmed cell death caused by defects in NMD, but not additional cell cycle defects, as implied by the incomplete suppression in the *Drosophila* eye and mammalian cell culture. Such defects in the cell cycle may be particularly pronounced during the development of certain tissue, or specific developmental stages. Indeed, NMD has been reported to have differing stage and tissue- specific activities (*Bao et al., 2015*; *Bruno et al., 2011*; *Colak et al., 2013*; *Li et al., 2015*). Whether this is due to a role in surveillance or another specific target remains unclear, but examination of the effects of loss of NMD in *Gadd45* mutants should allow exploration of these possibilities.

The benefit for such a mechanism regulating *Gadd45* expression may lie in a function of NMD in restricting viral growth (*Balistreri et al., 2014*). Because viruses encode *trans*-acting factors to inhibit NMD (*Mocquet et al., 2012*), the resulting accumulation of GADD45 in infected cells may act as a

"molecular tripwire" that rapidly elicits a stress response and cell death. This outcome suggests that regulating responses to infection may underlie a conserved essential function of NMD. Intriguingly, restriction of pathogens via NMD extends to plants (*Garcia et al., 2014*), where NMD mutant lethality in *A. thaliana*, which do not encode *Gadd45* orthologues, may be caused by the overexpression of a subset of immune-related intracellular nucleotide-binding leucine-rich repeat receptors, some of which are endogenous NMD targets (*Gloggnitzer et al., 2014*). In contrast, eukaryotes that do not rely on the activation of programmed cell death to protect against viruses, such as *S. cerevisiae, S. pombe*, and *C. elegans*, do not require NMD for viability (*Hodgkin et al., 1989*; *Leeds et al., 1991*; *Mendell et al., 2000*). Together these observations suggest a potential novel role for NMD and *Gadd45* in immune responses, triggering the death of infected cells during pathogenic challenges.

Restoring the expression of PTC-containing alleles via NMD inhibition has been proposed as a promising therapy for a wide range of recessive genetic diseases (*Keeling et al., 2014*). Translation of stable PTC-containing mRNAs would produce truncated proteins that may be partially functional and alleviate disease symptoms normally caused by complete loss of the protein. However, the essential function for NMD in viability has raised the concern that these therapies may have prohibitive side effects. Our findings reveal a molecular basis for dealing with this obstacle by suggesting that inhibiting both the NMD and *Gadd45* pathways (*Tornatore et al., 2014*) in combination could provide an effective and safe treatment for patients with debilitating genetic disorders.

## Materials and methods

### Fly genetics

*Drosophila melanogaster* stocks were raised on standard cornmeal/dextrose food at 25°. The NMD mutant alleles $Upf2^{25G}$, $Upf2^{7-5A}$, and $Upf1^{26A}$ (*Frizzell et al., 2012*; *Metzstein and Krasnow, 2006*) are on $y\ w\ FRT^{19A}$ chromosomes. These alleles were balanced over *FM7i, P{ActGFP}JMR3* (*Reichhart and Ferrandon, 1998*). $Smg5^{G115}$ and $Smg5^{C391}$ are null alleles of *Smg5* (J.O.N., D. Förster, S. Luschnig, and M.M.M., unpublished) and will be described in detail later. The *Smg5* alleles are balanced over *CyO, P{Dfd:eYFP $w^+$}* (*Le et al., 2006*). Other alleles used were *P{w[+mC]=EPg} HP20647* (*Staudt et al., 2005*), $Mekk1^{Ur36}$ (*Inoue et al., 2001*) recombined on $FRT^{82B}$ by D. Ryoo, *ey-FLP* (*Newsome et al., 2000*), $pcm^{14}$ (*Waldron et al., 2015*), $Adh^{n4}$ (*Chia et al., 1987*) and $DHR78^3$ (*Fisk and Thummel, 1998*). Control chromosomes were $y\ w\ FRT^{19A}$ (for *Upf1* and *Upf2*) and $FRT^{82B}$ (for *Mekk1*) (*Xu and Rubin, 1993*). For all experiments using $Gadd45^{F17}$ we used the $Gadd45^{E8}$ precise excision as a control.

For viability assays, we mated flies for 3 days and collected all progeny each day for 10 days, starting 10 days after the cross was initiated. The total numbers of F1 mutant and balancer males were scored, and the ratio of mutant males to balancer males was used to determine mutant animal viability. To control for balancer viability within each experiment, we normalized the ratio of mutant to balancer animals to a ratio of the appropriate control chromosome to balancer animals produced from a parallel cross.

### Deficiency suppressor screen

We screened autosomal deficiencies from the DrosDel collection (*Ryder et al., 2007*). All deficiencies scored can be found in *Supplementary file 1*. Deficiencies on chromosome 2 were balanced over *CyO*, and deficiencies on chromosome 3 were balanced over *TM6C*. We mated males from each deficiency stock to $y\ w\ Upf2^{25G}\ FRT^{19A}$/*FM7i, P{ActGFP}JMR3* females and scored all F1 males for the presence or absence of each balancer. For any given deficiency tested, the percentage of *Deficiency / +* males that are $Upf2^{25G}$ mutants, less the percentage of *Balancer / +* males that are $Upf2^{25G}$ mutants was calculated, producing a Deficiency Suppression Score (DSS), which represents the effect of an individual deficiency on the increase or decrease in $Upf2^{25G}$ viability, while controlling for each deficiency's general influence on viability. A DSS greater than 0.1 indicates suppression of lethality. Supplemental deficiencies used were from the Exelixis collection (*Parks et al., 2004*) and *Df(2R)sple-J1* (*Heitzler et al., 1993*). Deficiency mapping to the *Drosophila* genome was performed using the 5.1 genome release.

RNA-seq data sets were acquired from *Chapin et al. (2014)* (archives SRR896609, SRR896616, SRR503415, and SRR503416) and aligned using Bowtie and TopHat alignment with standard

remapping parameters to the 5.1 *Drosophila* genome release. SAMtools accessory scripts were used to retrieve read counts for deficiency and control regions. All read counts were normalized to reads per million within each data set. Average normalized reads in $Upf2^{25G}$ samples were normalized to the relative reads of 74 ribosomal proteins in $Upf2^{25G}$ samples compared to control samples. Total normalized reads within the regions removed by each deficiency were averaged between biological replicates, and the difference between the $Upf2^{25G}$ and control samples was divided by one million to determine percent increase in genomic load across each deficiency region.

## Generation of *Gadd45* mutants

We produced P-element excision lines from the *P{w[+mC]=EPg}HP20647* P-element insertion line crossed to a *Δ2–3* transposase stock. We mated F1 males containing the P-element and transposase on a *CyO* balancer to *w; Tft / CyO* females. *Cy⁺ Tft* white-eyed F2 males were then individually mated to *w; Tft / CyO* females. We then collected *Tft⁺, Cy* males and females to create an isogenic stock from each individually mated F2 male. To identify precise excisions we used the primers Gadd45_F1 / Gadd45_R1 flanking the P-element insert site to amplify a region across the excised P-element. Lines that failed to amplify with these primers were candidate imprecise excisions, which we then tested with Gadd45_F1 / Gadd45_R3 primers for deletions. Any detected deletions were subsequently sequenced using these same primers. Primer sequences are found in *Supplementary file 2*.

## Induction and analysis of eye clones

We generated eye clones with the FLP/FRT system using the *ey-FLP* driver (*Newsome et al., 2000*) to induce recombination. We imaged eyes on a Leica MZ125 stereo microscope with a Retiga-2000R camera (QImaging, Canada) with QCapture 3.1.2 software (QImaging). We focused images using the ImageJ stack focuser plugin and quantified relative eye clone size using the ImageJ analyzer tools. A total of 20 eyes from 20 individual animals were scored for each condition.

## Cell death assays

For TUNEL assays, third instar larval wing discs were dissected as described in Sullivan *et al.* (*Sullivan et al., 2000*). TUNEL staining was performed using the Apoptag Red in situ Apoptosis Detection Kit (Chimicon International Inc., Billerica, MA) according to Chakraborty *et al.* (*Chakraborty et al., 2015*). We DAPI stained wing discs (1:5000) for 5 min prior to mounting. Confocal images were acquired using a Zeiss LSM710 laser scanning confocal microscope (Carl Zeiss AG, Germany). 3-dimensional datasets were acquired with a Plan-Apochromat 20X/0.8 lens, 1.34 μm z-step, using the Zeiss ZEN software. To measure TUNEL signal intensity z-projections images were summed with ImageJ. Background signal was removed by using the ImageJ MaxEntropy auto-threshold. Relative total TUNEL signal intensity was calculated using the ImageJ analyzer tools to measure the total pixel intensity within the wing discs of TUNEL images and normalized to the average intensity in control conditions.

For annexin V staining, we collected media (including floating cells) from siRNA treated cells. We spun down media at 950g for 4 min to pellet cells, and then aspirated remaining media. Concurrently, we trypsinized siRNA-treated cells still on plates and added them to the same respective tube as previously spun-down media. Following the Alexa Fluor 488 Annexin V/Dead Cell Apoptosis Kit (Abcam, UK) protocol, we stained for apoptotic cells. We visualized cells on an Olympus IX51 microscope (Olympus, Japan) with 20X objective. We collected bright field as well as fluorescent images using a FITC filter with a QImaging QICam Fast1394 camera and QCaptureP software (QImaging). We analyzed cells by counting all cells within a bright field image as well as the annexin V positive cells from the same image. The number of annexin V positive cells was divided by total cell number to generate the fraction of apoptotic cells for each treatment. >3000 total cells were counted across three biological replicates for each treatment.

## Cell culture experiments

We cultured mouse NIH-3T3 cells (ATCC) or HEK293 cells (ATCC) in DMEM (Thermo-Fisher, Waltham, MA) supplemented with 10% fetal bovine serum and glutamine. For siRNA experiments, we transfected cells using RNAiMax and 24 pmol of negative control siRNA

(Qiagen, Netherlands), *Upf1* siRNA (Qiagen), or *Gadd45b* siRNA (Sigma-Aldrich) for 3T3 cell experiments, or negative control siRNA (Qiagen), *UPF1* siRNA (Qiagen), or *GADD45A* siRNA (Sigma-Aldrich, St. Louis, MO) for HEK293 experiments. For double siRNA-treated cells, we used 24 pmol of each *Upf1* and *Gadd45b* siRNA for 3T3 experiments or *UPF1* and *GADD45A* siRNA for HEK293 experiments.

For actinomycin experiments, we incubated cells with siRNA for 48 hr before changing the media and then incubated with 2 µg/mL actinomycin (Sigma-Aldrich) for 1 or 2 hr. mRNA half-life was determined by fitting an exponential decay curve to the relative expression at each time point (*Tani et al., 2012*). $t_{1/2}$ was calculated based on the average expression at each time point, and the mean $t_{1/2}$ for each condition is represented.

For cell counting experiments, we trypsinized cells, incubated a small aliquot with Trypan Blue at a final concentration of 0.04% in complete media, and counted Trypan Blue negative cells. RNA was collected from the remaining cells, and relative mRNA levels were measured as described below.

## RNA isolation and quantitative RT-PCR

For *Drosophila* qRT-PCR analyses, we collected 5–10 adult animals frozen in liquid nitrogen. We isolated total RNA using TRIzol reagent (Invitrogen) and phase-lock tubes (5-Prime), and the RNeasy mini kit (Qiagen). We used on-column RNase-free DNase treatment (Qiagen) to reduce genomic contamination. We determined RNA concentration by spectrophotometer and normalized concentration for reverse transcription. For reverse transcription, we used random decamers and MMLV8 reverse transcriptase (Retroscript Kit, Thermo-FIsher). We performed qRT-PCR analysis using the SYBR Green qPCR Supermix (Bio-Rad, Hercules, CA) and the Bio-Rad iCycler thermocycler. All experimental reactions were performed using three technical replicates and a minimum of three biological replicates per condition, and the expression level of all experimental assays was normalized to *RpL32* mRNA expression.

For cell culture qRT-PCR analyses, we collected RNA following the Zymo Research Quick RNA MiniPrep kits protocol, and synthesized cDNA using MMLV reverse transcriptase (NEB, Ipswich, MA) with a template of 1 µg of total RNA and priming with a T18 oligo. We measured relative mRNA levels by qRT-PCR using the Masterplex ep realplex (Eppendorf, Germany) with SYBR green fluorescent dye. Each sample was measured with technical triplicates and three biological replicates, and target mRNA levels were normalized to those of ribosomal protein 19 (*Rpl19*) mRNA.

For all qRT-PCR analyses we also measured samples that had been made without reverse transcriptase to ensure that signal was not due to genomic DNA. Primer sequences can be found in *Supplementary file 2*.

## 3′ UTR cloning and sensitivity assay

We cloned the *UAS-GFP::Gadd45 3′ UTR* and control *UAS-GFP::Act5C 3′ UTR* constructs using the primers G45_3U_X1_F / G45_3U_S1_R or Act5C_X1_F / Act5C_S1_R (*Supplementary file 2*) to amplify the *Gadd45* and *Act5C* 3′ UTRs, respectively, from genomic DNA. PCR fragments were inserted into the Zero Blunt TOPO vector (Thermo-Fisher), sequenced to assure fidelity, and digested and cloned into a pUAST-attB GFP vector using standard cloning procedures to replace the SV40 3′ UTR. Plasmids were injected by BestGene (Chino Hills, CA) into a stock containing the VK00027 attP site (*Venken et al., 2006*) for *phiC31* directed integration. We used previously described *UAS-GFP::SV40 3′ UTR* animals (*Metzstein and Krasnow, 2006*). For imaging, wandering late L3 larvae were collected and examined using a Leica MZ 16F microscope and the Leica DFC340 FX camera with the Leica Application Suite v3.3.0 software.

## Analysis of *dHR78³* and *Adhⁿ⁴* PTC allele stability

We collected adult F1 *Upf2⁺; Gadd45^{E8/+}*, *Upf2^{25G}; Gadd45^{E8/+}*, and *Upf2^{25G}; Gadd45^{F17/+}* males that were also heterozygous for either the *dHR78³* or *Adhⁿ⁴*. The *Adhⁿ⁴* allele is a PTC-containing allele and has been demonstrated to be a direct NMD target based on cleavage by Smg6 (*Gatfield and Izaurralde, 2004*). The *dHR78³* allele is also a PTC-containing allele and thus is presumably degraded by NMD (*Fisk and Thummel, 1998*). At least three biological replicates were collected for each condition. We isolated RNA and generated cDNA as described in methods above and used this cDNA as a template for PCR amplification of the *dHR78* transcript with the DRH78_F3

/ DHR78_R3 primers and the *Adh* transcript with the Adh_F and Adh_R primers (*Supplementary file 2*), which flank the nonsense mutation in the respective transcripts. To compare the relative abundance of the *dHR78³* allele to the wild-type allele, PCR products were Sanger sequenced, and the relative peak intensity for a T (*dHR78³* allele) compared to a C (wild-type allele) at nucleotide 1063 was compared. To compare the relative abundance of the *Adh^{n4}* allele to the wild-type allele, PCR products were digested with *Pvu*II (a site disrupted by the *n4* mutation), separated on a 1% agarose gel and stained with ethidium bromide. The relative intensity of the cut and uncut bands was determined using ImageJ and normalized for fragment length. All samples were ran on the same gel and compared under identical conditions. All ratios were normalized to the ratio in the *Upf2^{25G}*; *Gadd45^{E8/+}* condition.

## Statistical analysis

All figures displaying viability assays represent a proportion of animals of the indicated genotypes that survive to adulthood; error bars for these figures represent the 95% confidence interval of the binomial distribution, and the Test of qual or Given Proportions was used to determine significance difference in these proportions between genotypes. All other figures represent the mean value of multiple replicates have error bars depicting $\pm 2$ SEM, which is a close approximation of the 95% confidence interval (*Krzywinski and Altman, 2013*). For tests between two variable measures, a two-sided paired Student's t-test was used to determine significance difference between mean value data. For most qPCR experiments, data was compared to a normalized control, set to a constant of 1, so these tests were performed with a one-sided Student's t-test.

## Acknowledgements

We thank the Metzstein and Thummel labs for helpful discussion; Shawn Rynearson, Esther Ellison, Zev Kronenberg, and EJ Osborne for assistance with collection and analysis of the deficiency suppressor screen; Kate Sanders for assistance isolating the *Gadd45^{F17}* allele; Don Ryoo, Stefan Luschnig, Sarah Newbury, and Carl Thummel for providing *Drosophila* lines; Kim Frizzell for assistance imaging *Drosophila* eyes; Ria Chakraborty and Kent Golic for assistance with TUNEL; Chase Bryan for assistance with confocal microscopy; and Carl Thummel, Gillian Stanfield, and Nels Elde for helpful comments on the manuscript. Fly stocks were obtained from the Bloomington *Drosophila* Stock Center. Exelixis deficiencies were provided by Exelixis, Inc. This work was supported by National Institutes of Health (NIH) grant 1R01GM084011 (to MMM) and a March of Dimes Award 5-FY07-664 (to MMM).

## Additional information

### Funding

| Funder | Grant reference number | Author |
|---|---|---|
| National Institutes of Health | 1R01GM084011 | Jonathan O Nelson Mark M Metzstein |
| March of Dimes Foundation | 5-FY07-664 | Mark M Metzstein |

The funders had no role in study design, data collection and interpretation, or the decision to submit the work for publication.

### Author contributions

JON, MMM, Conception and design, Acquisition of data, Analysis and interpretation of data, Drafting or revising the article; KAM, JH, Acquisition of data, Analysis and interpretation of data, Drafting or revising the article; AC, Conception and design, Analysis and interpretation of data, Drafting or revising the article, Contributed unpublished essential data or reagents

### Author ORCIDs

Jonathan O Nelson, http://orcid.org/0000-0001-9831-745X
Mark M Metzstein, http://orcid.org/0000-0002-4105-2750

## Additional files

### Supplementary files

• Supplementary file 1. Deficiencies used in deficiency suppressor screen.

• Supplementary file 2. PCR primers used in this study.

### Major datasets

The following previously published dataset was used:

| Author(s) | Year | Dataset title | Dataset URL | Database, license, and accessibility information |
|---|---|---|---|---|
| | 2014 | D.melanogaster Nonsense-Mediated mRNA Decay study | http://trace.ncbi.nlm.nih.gov/Traces/sra/?study=SRP025939 | http://trace.ncbi.nlm.nih.gov/Traces/sra/ |

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
