## [Decision Letter]

Thank you for submitting your work entitled "Degradation of a single mRNA by nonsense-mediated decay is essential for viability" for consideration by *eLife*. Your article has been reviewed by three peer reviewers, one of whom is a member of our Board of Reviewing Editors, with James Manley as the Senior Editor.

The reviewers have discussed the reviews with one another and the Reviewing Editor has drafted this decision to help you prepare a revised submission.

Summary:

A critical question in the RNA field is the functional significance of the nonsense-mediated RNA decay (NMD) pathway. NMD was originally discovered on the basis of its ability to degrade aberrant mRNAs (i.e., its RNA surveillance role) but was subsequently found to degrade many normal mRNAs (i.e., its gene expression role). Which is more important? With this elegant study, Mark Metzstein and colleagues take a great leap towards answering this question.

The authors sought to identify abnormally stabilized mRNAs that mediate lethality in NMD-defective *Drosophila melanogaster*. To this end, a genetic suppressor screen was performed in a *Drosophila* NMD mutant strain (hypomorphic *Upf2*^25G^ allele) and three candidate suppressing regions were identified. Two of these encompass the *Gadd45* and *Mekk1* genes, which operate in the same stress-response pathway. The importance of these genes for the NMD-lethality phenotype was analyzed further, and it was demonstrated that depletion of *Gadd45* in the *Upf2*^25G^-mutant completely restored viability, whereas it partially rescued viability in NMD null flies. Depletion of *Mekk1*, which acts downstream of *Gadd45* in the stress response, also increased viability, but to a lesser degree. Next, it was demonstrated that *Gadd45* is a direct target of NMD and that increased levels of *Gadd45* can explain the increased cells death observed in the *Upf2*^25G^-mutant.

Finally, the authors provided evidence that *Gadd45b* mRNA is also a direct NMD target in mouse NIH-3T3 cells and that its increased expression can partially explain the decreased viability observed in *UPF1*-depleted cells.

These observations led the authors to conclude that the essentiality of NMD in *Drosophila* and mammals in essence resides in the down-regulation of a single gene (*Gadd45*), which is involved in induction of apoptosis. The presented results, coupled with the finding of others that viral infection suppresses NMD, made the authors propose an interesting model in which NMD provides defense against virus infection by killing virally infected cells.

Essential revisions:

Overall, this study represents an important advance for the RNA/NMD field. Publication in *eLife* is therefore suggested if the authors can substantiate their conclusions. In its present form, the main conclusion (including the title) appears to be going somewhat too far. Several of the authors' own observations point to additional contributions to lethality besides higher *Gadd45* expression – except in the specific *Upf2*^25G^-mutant.

1) It is important to prove that the observed effects are coupled to increased *Gadd45* protein. The authors need to show that expression of the *Gadd45* protein is increased in *Upf2*^25G^ and perhaps also other NMD-defective cells.

2) Figure 2. It is critical that the authors examine suppression of apoptosis by *Gadd45* in *Upf1*-mutant flies (e.g., *Upf1^26A^*), not just *Upf2*-mutant flies (as both were examined in Figure 1). Given that NMD factors can function in pathways other than NMD, it is essential to assess more than one NMD factor.

3) Conservation in mammals. The results in Figure 3 are intriguing, but require some bolstering.

A) The mechanism underlying the alterations in cell counts must be investigated. Is it the result of alterations in cell death, proliferation, or both? Is the *Gadd45*-dependent apoptosis phenotype conserved in mammals?

B) At least one more cell line should be used for at least some experiments (one cell line is not enough to be convincing).

C) Given that there are several *Gadd45* genes in mammals, it is important that these also be investigated. While it is appreciated that *Gadd45b* is the most highly expressed *Gadd45* gene in NIH-3T3 cells, this does not rule out the importance of the other *Gadd45* genes. Perhaps try the triple knockdown of *Gadd45A, B* and *G. GADD45A* appears to be the highest expressed variant in HEK293 and HeLa cells, so one or both of these cell lines could be obvious additional cell lines to test.

D) Figure 3. One time point is not sufficient for RNA half-life analysis.

E) Figure 3. More NMD substrates besides *Smg5* should be used for this analysis. Indeed, the effect on *Smg5* is very modest (probably because of either insufficient NMD factor knockdown or poor transfection efficiency). A lack of an effect on some NMD substrates is ok given the heterogeneity of the NMD pathway, but at least one NMD substrate should have a greater effect than only 50%. A cell line with greater transfection efficiency than NIH-3T3 might be necessary to use. The GAS5 non-coding RNA could be a possible candidate to check since it appears to be highly responsive to NMD in both human and mouse cells.

4) Although it is very clear that *Gadd45* stabilization is a major determinant for the lethality phenotype of the *Upf2*^25G^-mutant *Drosophila* strain and the decreased viability phenotype in *UPF1*-depleted NIH-3T3 cells, it is also clear that increased levels of *Gadd45* cannot fully explain the lethality of NMD null mutant flies. Therefore, there must be other transcripts at play. One possibility is that NMD is vital for different stages of development, as e.g. indicated by several NMD knockout mouse studies. The authors should discuss more thoroughly the possibility that different stages of development may be affected by abnormal high expression of other factors than *Gadd45*. Generally, the conclusions (incl. the title) should be somewhat toned down, because overexpression of *Gadd45* cannot fully explain lethality of all NMD-mutants. In relation to this: does the third deleted region contain genes that are related to the *Gadd45* pathway or are they unrelated? In case of the latter, this also points to other contributors to lethality.

---

## [Author Response]

*1) It is important to prove that the observed effects are coupled to increased Gadd45 protein. The authors need to show that expression of the Gadd45 protein is increased in Upf2^25G^ and perhaps also other NMD-defective cells.*

We could not directly address Gadd45 protein expression in *Upf2^25G^* flies because no antibodies that suitably detect *Drosophila* Gadd45 protein exist. We attempted to detect *Drosophila* Gadd45 using a broadly reacting antisera directed against HumanGADD45β, however, we found that this antibody does not cross-react with *Drosophila* Gadd45 protein. In addition, we tried to use this antisera in our mammalian cell culture, and also could not detect expression of endogenous Gadd45b (in the literature developed antibodies have only been tested on cells over-expressing Gadd45). However, since our data indicates that the loss of *Mekk1* shows similar effects to loss of *Gadd45* and that Gadd45 is known to regulate *Mekk1* by a protein-protein interaction strongly suggests that Gadd45 protein levels must be increased in NMD mutants, even in the absence of a direct assay of Gadd45 protein levels in *Drosophila*. Furthermore, we show that a GFP reporter with the *Gadd45* 3’ UTR has increased fluorescence in *Upf2^25G^* mutants, also suggesting that transcripts with the *Gadd45* 3’UTR have increased protein levels in NMD mutants.

*2) Figure 2. It is critical that the authors examine suppression of apoptosis by Gadd45 in Upf1-mutant flies (e.g., Upf1^26A^), not just Upf2-mutant flies (as both were examined in Figure 1). Given that NMD factors can function in pathways other than NMD, it is essential to assess more than one NMD factor.*

We agree with the reviewers that it is important to test if loss of *Gadd45* suppresses cell death in other NMD mutants, such as *Upf1* or *Upf2* null alleles. However, we have previously found that such mutants die during early larval stages(Chapin et al. 2014) and we were not able to isolate discs from these animals to examine cell death. Instead, we turned to another critical NMD gene, *Smg5*. In work we are currently preparing for publication we show that *Drosophila Smg5* is critical for NMD function and viability, similar to *Upf1* and *Upf2*, but *Smg5* mutants die at a later stage. We examined TUNEL signal in the wing discs of *Smg5* null mutants and found it is increased compared to controls and that *Gadd45* mutants suppressed this ectopic cell death, very similarly to what we found in *Upf2^25G^* mutant discs. This result indicates that the cell death in *Upf2* mutant wing discs is in fact a defect due to loss of NMD pathway function and not the loss of NMD-independent *Upf2* function. We are presenting this data in Figure 2—figure supplement 1. As a technical note, when analyzing the *Smg5* wing discs our reagents lead to higher background signal than the previous experiments, so we had to adapt our quantification procedure (as described in the Methods section). In the interest of consistency, we reanalyzed our *Upf2^25G^* data using this new approach and obtained essentially identical results to before. We have updated Figure 2 to represent this new analysis.

*3) Conservation in mammals. The results in Figure 3 are intriguing, but require some bolstering.*

*A) The mechanism underlying the alterations in cell counts must be investigated. Is it the result of alterations in cell death, proliferation, or both? Is the Gadd45-dependent apoptosis phenotype conserved in mammals?*

We have examined the mechanism for cell count changes by staining cells for the apoptotic marker annexin V. We find that Upf1 knockdown cells show a large increase in apoptosis, and this increase is almost entirely suppressed by simultaneous knockdown of *Gadd45b*. However, the increase in cell death probably does not alone account for the decrease in cell numbers, suggesting an underlying proliferation defect also occurs in NMD knockdown. Since *Gadd45b* knockdown appears to totally eliminate the apoptosis defect in NMD knockdown cells, but does not fully rescue cell numbers, this further suggests *Gadd45b* overexpression may not be responsible for the proliferation defect. We come to similar conclusions from *Drosophila* eye analysis. We have included these data in Figure 3, and in the Discussion (this is also relevant to comment 4, below)

*B) At least one more cell line should be used for at least some experiments (one cell line is not enough to be convincing).*

We have repeated our experiments using human HEK293 cells and obtained very similar results. These are shown in Figure 3—figure supplement 1.

*C) Given that there are several Gadd45 genes in mammals, it is important that these also be investigated. While it is appreciated that Gadd45b is the most highly expressed Gadd45 gene in NIH-3T3 cells, this does not rule out the importance of the other Gadd45 genes. Perhaps try the triple knockdown of Gadd45A, B and G. GADD45A appears to be the highest expressed variant in HEK293 and HeLa cells, so one or both of these cell lines could be obvious additional cell lines to test.*

We have attempted multiple Gadd45 knockdowns in both 3T3 and HEK cells, but we have found these to have deleterious effects on cell survival on their own, greatly convoluting our analysis of suppression of NMD knockdown. In these experiments we did find, as predicted by the reviewers, that GADD45A seems to primarily mediate the NMD knockdown response in HEK cells, so it is indeed the case that the predominantly expressed Gadd45 that seems to mediate NMD knockdown effects. However, since we show that we can rescue NMD cells by knockdown of a single homolog in both cell types, we believe the conclusions we present in the manuscript are valid. Knockdown of multiple Gadd45 paralogs, if it were possible, would be expected to further improve NMD cell survival, strengthening our conclusion, but the negative result of no increased survival would not suggest an alternative model.

*D) Figure 3. One time point is not sufficient for RNA half-life analysis.*

We have added an additional time point and used this to calculate RNA half-life directly. This is presented as a revised panel in Figure 3.

*E) Figure 3. More NMD substrates besides Smg5 should be used for this analysis. Indeed, the effect on Smg5 is very modest (probably because of either insufficient NMD factor knockdown or poor transfection efficiency). A lack of an effect on some NMD substrates is ok given the heterogeneity of the NMD pathway, but at least one NMD substrate should have a greater effect than only 50%. A cell line with greater transfection efficiency than NIH-3T3 might be necessary to use. The GAS5 non-coding RNA could be a possible candidate to check since it appears to be highly responsive to NMD in both human and mouse cells.*

We have replaced the weakly affected *Smg5* with two other substrates whose RNA levels are more than two-fold affected by NMD knockdown. We would have liked to have tested GAS5 directly, but all our current samples are polyA selected, so we unfortunately do not have this RNA represented.

4) Although it is very clear that Gadd45 stabilization is a major determinant for the lethality phenotype of the Upf2^25G^-mutant Drosophila strain and the decreased viability phenotype in UPF1-depleted NIH3T3 cells, it is also clear that increased levels of Gadd45 cannot fully explain the lethality of NMD null mutant flies. Therefore, there must be other transcripts at play. One possibility is that NMD is vital for different stages of development, as e.g. indicated by several NMD knockout mouse studies. The authors should discuss more thoroughly the possibility that different stages of development may be affected by abnormal high expression of other factors than Gadd45. Generally, the conclusions (incl. the title) should be somewhat toned down, because overexpression of Gadd45 cannot fully explain lethality of all NMD-mutants. In relation to this: does the third deleted region contain genes that are related to the Gadd45 pathway or are they unrelated? In case of the latter, this also points to other contributors to lethality.

The reviewers make valid points about the likelihood of additional factors contributing to lethality in NMD mutant animals. We have incorporated these suggestions into the Discussion, and modified our title to “Degradation of *Gadd45* mRNA by nonsense-mediated decay is essential for viability”, which more explicitly refers to the necessity of *Gadd45* regulation by NMD. In addition, our data suggests that loss of NMD causes defects in cell survival via GADD45 upregulation, but also defects in the cell cycle independent of Gadd45. These latter defects do not always lead to cell lethality, but may contribute to the lower viability observed in NMD null mutants even with *Gadd45* deleted. These points have been added to the Discussion.

Preliminary analysis of the third region indicates that suppression is likely due to multiple loci, and it is unclear if these loci function on the *Gadd45* pathway (like *Mekk1*) or are independently regulated by NMD. We are continuing to analyze genes in this region and will report on it in a future publication.